# Generalization of the Change of Variables Formula
# with Applications to Residual Flows

**Niklas Koenen** [1 2]  **Marvin N. Wright** [1 2]  **Peter Maaß** [1]  **Jens Behrmann** [1]

## Abstract

Normalizing flows leverage the Change of Variables Formula (CVF) to define flexible density models. Yet, the requirement of smooth transformations (diffeomorphisms) in the CVF poses a significant challenge in the construction of these models. To enlarge the design space of flows, we introduce $\mathcal{L}$-diffeomorphisms as generalized transformations which may violate these requirements on zero Lebesgue-measure sets. This relaxation allows e.g. the use of non-smooth activation functions such as ReLU. Finally, we apply the obtained results to planar, radial, and contractive residual flows.

## 1. Introduction

The term normalizing flow refers to a concatenation of arbitrarily many simple transformations such that together they describe a transformation of desired flexibility and expressiveness. Formally, a transformation $f : Z \rightarrow X$ denotes a *diffeomorphism*, i.e., a bijective mapping where both $f$ and $f^{-1}$ are continuously differentiable. The crucial reason why transformations are considered in normalizing flows is the validity of the *Change of Variables Formula (CVF)* for a probability density $p_Z$ on $Z$ described by

$$\int_{f^{-1}(A)} p_Z(z) \, d\lambda^n(z) = \int_A p_f(x) \, d\lambda^n(x), \quad (1)$$

where $p_f(x) := p_Z(f^{-1}(x))|\det J_{f^{-1}}(x)|$ and $A \subseteq X$. This formula provides an explicit expression of the density $p_f$ induced by $f$ on the target space $X$, which includes the determinant of the Jacobian as a volume correction term.

Based on this definition of a transformation, however, it is generally not accurate to use non-smooth activations,

---

[1]Faculty of Mathematics and Computer Science, University of Bremen, Bremen, Germany [2]Leibniz Institute for Prevention Research and Epidemiology – BIPS, Bremen, Germany. Correspondence to: Niklas Koenen <koenen@leibniz-bips.de>.

Third workshop on *Invertible Neural Networks, Normalizing Flows, and Explicit Likelihood Models* (ICML 2021). Copyright 2021 by the author(s).

such as ReLU, Leaky ReLU, or ELU with $\alpha \neq 1$, in the design of normalizing flows. These usually cause the flow to become non-differentiable on a set with no volume w.r.t. the Lebesgue measure $\lambda^n$, hence no diffeomorphism. In measure theory, these sets are called $\lambda^n$-*null sets* or only *null sets* for short and are negligible in integration.

We demonstrate that the requirements for a flow can be significantly weakened by excluding null sets from the base and target space while preserving the validity of the CVF. There are remarks on using almost everywhere (a.e.) differentiable activation functions in Kobyzev et al. (2020) or Kong & Chaudhuri (2020), yet both works lack a proof of the validity of such transformations to define flows. In our work, we provide such proofs for an even more general statement. At the same time, we discuss the probabilistic background of normalizing flows to induce a well-defined density in the end. Furthermore, we point out that not every generalization of the CVF found in the mathematical literature is immediately suitable for flows. Finally, we put a special emphasis on the applications to residual flows. In doing so, we prove that non-smooth activations are also valid for both planar and radial flows (Rezende & Mohamed, 2015), as well as for contractive residual flows (Behrmann et al., 2019).

## 2. Background on CVF in Probability Theory

The basic idea behind normalizing flows is to transform a known and tractable probability space into a more complex one. Mathematically, a probability space $(Z, \mathcal{A}_Z, \mathbb{P}_Z)$ is composed of a set $Z$ equipped with a $\sigma$-algebra and a probability measure $\mathbb{P}_Z$. In the target space $(X, \mathcal{A}_X, \mathbb{P}_{\text{data}})$, only the set and $\sigma$-algebra are fixed, and the data distribution $\mathbb{P}_{\text{data}}$ is unknown. For simplicity, we only consider open subsets of $\mathbb{R}^n$ and trace $\sigma$-algebras of the Lebesgue algebra $\mathcal{L}$ in the following, i.e., $\mathcal{A}_Z = \mathcal{L}(Z)$ and $\mathcal{A}_X = \mathcal{L}(X)$. The *trace $\sigma$-Algebra* is a restricted $\sigma$-Algebra on a subset defined by $\mathcal{L}(Z) := \{A \cap Z \mid A \in \mathcal{L}\}$. Besides, we assume that the distribution $\mathbb{P}_Z$ is absolutely continuous w.r.t. the $n$-dimensional Lebesgue measure $\lambda^n$, i.e., $\lambda^n$-null sets have a $\mathbb{P}_Z$-probability of zero. Therefore, the existence of a probability density $p_Z$ follows by Radon-Nikodym's theorem (Bogachev, 2006, Sec. 3.2). For more information on

measure theory, see Bogachev (2006) or Elstrodt (2013).

At the lowest level, a transformation $f : Z \to X$ has to be at least an $\mathcal{A}_Z$-$\mathcal{A}_X$-measurable mapping (i.e., a random variable) in order to induce a distribution on the target space by the so-called *pushforward measure*:

$$\mathbb{P}_f(A) := \mathbb{P}_Z\left(f^{-1}(A)\right) \qquad (A \in \mathcal{A}_X).$$

Under the assumption that the base distribution $\mathbb{P}_Z$ has a density function $p_Z$, there also exists an integral representation for the pushforward measure, i.e.,

$$\mathbb{P}_f(A) = \int_{f^{-1}(A)} p_Z(z)\, d\lambda^n(z). \tag{2}$$

**Assumption 1** (Transformations for the CVF). The function $f : Z \to X$ between two open sets $Z, X \subseteq \mathbb{R}^n$ is a diffeomorphism; or equivalently expressed by the inverse function theorem, $f$ is bijective, continuously differentiable and without critical points.

We note that $z \in Z$ is a *critical point* if the Jacobian-determinant vanishes in this point, i.e., $\det J_f(z) = 0$. In particular, a critical point $z$ indicates that the inverse is non-differentiable or non-continuously differentiable in $f(z)$.

If the mapping $f$ satisfies Assumption 1, the CVF from eq. (1) holds, and we can extend the expression (2) of the distribution $\mathbb{P}_f$ by

$$\mathbb{P}_f(A) = \int_{f^{-1}(A)} p_Z(z)\, d\lambda^n(z) = \int_A p_f(x)\, d\lambda^n(x). \tag{3}$$

Since the equality (3) is valid for all $\mathcal{A}_X$-measurable sets, the $\lambda^n$-unique probability density of $\mathbb{P}_f$ is given by

$$p_f(x) = p_Z\left(f^{-1}(x)\right)\left|\det J_{f^{-1}}(x)\right| \quad \text{(a.e. } x \in X). \tag{4}$$

In order to unify the existing definitions of flows in the literature, we will speak of a *flow* or a *proper flow* when a density on $X$ of the form as in eq. (4) is induced.

# 3. Generalization of the CVF

If it is argued that a flow indeed induces a probability density, often the CVF is merely named, and only in rare cases reference is made to sources like Rudin (1987) or Bogachev (2006). In most of the mathematical literature, it is proved in the following form

$$\int_{f(Z)} \psi(x)dx = \int_Z \psi(f(z))|\det J_f(z)|dz, \tag{5}$$

where $f : U \to \mathbb{R}^n$ is injective, continuously differentiable with $U \subseteq \mathbb{R}^n$ open, $Z \subset U$ measurable, and $\psi$ Lebesgue integrable. Moreover, there are even broader formulations

of this statement where $f$ is only differentiable or even only Lipschitz continuous everywhere and injective almost everywhere (cf. Varberg, 1971). In Bogachev (2006, Thm. 5.8.30) and Hajłasz (1993), generalizations are discussed where injectivity is not even required by considering the cardinality of the preimage set.

The identity (5) is, however, rather analytically motivated for solving integrals and does not aim to provide a representation of the density induced by $f$. In the common case where $f$ satisfies Assumption 1, these two variants (eq. (1) and (5)) are valid, since both $f$ and $f^{-1}$ form diffeomorphisms. Nevertheless, when we consider generalizations, it is usually no longer clear how and whether they can also be applied for the purposes of normalizing flows. In the following sections, we derive a similar strong generalization of the CVF as in (5), which is more suited to transforming probability densities, i.e., more suited for normalizing flows.

### 3.1. $\mathcal{L}$-Diffeomorphism

The basic idea is to require the conditions for a diffeomorphism only almost everywhere, since these sets do not affect integration. This idea leads to the following definition of a generalized transformation, providing a weaker set of conditions than in Assumption 1.

**Definition 2** (Lebesgue-Diffeomorphism). A mapping $f : Z \to X$ between two open sets $Z, X \subseteq \mathbb{R}^n$ is called *Lebesgue-diffeomorphism* ($\mathcal{L}$-*diffeomorphism* for short), if there are $\lambda^n$-null sets $N_Z, N_X$ with $N_Z$ closed such that the restriction $f : Z \setminus N_Z \to X \setminus N_X$ is bijective, continuously differentiable, and the set of critical points is a null set.

**Examples.** In the following, we list a few $\mathcal{L}$-diffeomorphism and their corresponding null sets:

1. A diffeomorphism $f : Z \to X$ forms an $\mathcal{L}$-diffeomorphism where both $N_Z$ and $N_X$ are empty sets.
2. The cubic function $f : \mathbb{R} \to \mathbb{R}$ with $f(x) = x^3$ is an $\mathcal{L}$-diffeomorphism with $N_Z = N_X = \emptyset$ and one critical point $\{0\}$, which is a null set.
3. In particular, the plane polar coordinates transformation $f : \mathbb{R}_+ \times [0, 2\pi] \to \mathbb{R}^2$ given by $f(r, \phi) = (r\cos(\phi), r\sin(\phi))$ is an $\mathcal{L}$-diffeomorphism with

$$N_Z = \{0\} \times [0, 2\pi] \cup \mathbb{R}_{>0} \times \{0, 2\pi\} \quad \text{and}$$
$$N_X = \mathbb{R}_+ \times \{0\}.$$

In appendix Lemma A.1 we show that a $\mathcal{L}$-diffeomorphism is a measurable mapping with respect to the corresponding trace $\sigma$-algebras of the Lebesgue algebra, thus inducing a distribution on $X$ by the pushforward measure. Moreover, the name 'diffeomorphism' does justice to the Definition 2, which we state in the following lemma (see Appendix A for the proof):

**Lemma 3.** *Let $f : Z \to X$ be an $\mathcal{L}$-diffeomorphism. Then there are $\lambda^n$-null sets $N_Z, N_X$ with $N_Z$ closed such that the restriction $f : Z \setminus N_Z \to X \setminus N_X$ is a diffeomorphism.*

The reasoning behind the assumptions of an $\mathcal{L}$-diffeomorphism can be understood as follows: We can remove negligible sets from the domain and the target space such that the restriction is bijective and continuously differentiable. To show that the restricted inverse is continuously differentiable, we use the inverse function theorem (Rudin, 1976, Thm. 9.24). Nevertheless, for the inverse function theorem to apply, all critical points and their image must still be removable, i.e., measure-theoretically negligible. That the set of critical points is a null set was assumed, and for the image, we use Sard's theorem (see Appendix A for the detailed proof).

Furthermore, it is necessary to suppose that the set of critical points is a null set; because there are examples of continuously differentiable and bijective functions whose set of critical points does not have measure zero. These points cause the inverse function to be non-continuously differentiable on a set with a positive measure (see Appendix A.2 for an example).

### 3.2. CVF for $\mathcal{L}$-Diffeomorphism

The previously defined $\mathcal{L}$-diffeomorphisms form a reasonable generalization of diffeomorphisms and are comparable to those transformations discussed in the introduction of this section. Furthermore, the following theorem justifies the validity of CVF as well for $\mathcal{L}$-diffeomorphisms (see Appendix A for the proof):

**Theorem 4** (CVF for $\mathcal{L}$-Diffeomorphism). *Let $f : Z \to X$ be an $\mathcal{L}$-diffeomorphism and $\mathbb{P}_Z$ a distribution on $Z$ with probability density $p_Z$ w.r.t. $\lambda^n$. Then the CVF (see eq. (3)) holds for $f$. In particular $f$ induces a distribution on $X$ with density $p_f$ given by*

$$p_f(x) = p_Z(f^{-1}(x)) \, |\det J_{f^{-1}}(x)| \quad (a.e. \ x \in X). \quad (6)$$

This theorem legitimizes the use of functions as proper flows that are not everywhere bijective, continuous, differentiable, or continuously differentiable. Even more, the inverse does not have to fulfill these properties everywhere either. In short, we can apply $\mathcal{L}$-diffeomorphisms as flows.

### 3.3. Invariance under Composition

The strength and tremendous upswing of normalizing flows mainly occurred because simple flows can be chained together. Thus, we can achieve the desired degree of complexity and expressiveness by increasing the number of simple flows. For this purpose, flows are often considered on the same base and target space. This crucial property is also retained for $\mathcal{L}$-diffeomorphisms (see Appendix A for the proof):

**Lemma 5** (Composition). *Let $\Omega \subseteq \mathbb{R}^n$ be an open set and $f_1, f_2 : \Omega \to \Omega$ $\mathcal{L}$-diffeomorphisms. Then the composition $f_2 \circ f_1$ is also an $\mathcal{L}$-diffeomorphism on $\Omega$.*

From this lemma, it results inductively that the concatenation $f = f_K \circ \ldots \circ f_1$ of $K$ $\mathcal{L}$-diffeomorphisms $f_1, \ldots f_K$ on $\Omega$ forms an $\mathcal{L}$-diffeomorphism. Hence, common formulas from the normalizing flow literature, e.g., presented in Kobyzev et al. (2020) or Papamakarios et al. (2021), also apply to $\mathcal{L}$-diffeomorphisms or can be extended to them. For example, the following holds for the Jacobian-determinant of $f$ with $x_i := f_{i+1}^{-1} \circ \ldots \circ f_K^{-1}(x)$ and $x_K := x$

$$\det J_{f^{-1}}(x) = \prod_{i=1}^{K} \det J_{f_i^{-1}}(x_i) \qquad (a.e. \ x \in \Omega).$$

Despite the legitimate use of $\mathcal{L}$-diffeomorphisms mathematically, it is essential to note that these can lead to numerical instabilities. In some points, the Jacobian-determinant or the inverse function does not need to exist. Nevertheless, these values can be set meaningfully or ignored in some situations.

## 4. Non-smooth Activations in Residual Flows

In this section, we apply the previous results to residual mappings, which are perturbations of the identity of the form $f(x) = x + g(x)$. This justifies the use of non-smooth activations in planar, radial, and contractive residual flows.

### 4.1. Planar Flows

The term *planar flow* was first introduced by Rezende & Mohamed (2015), which refers to functions of the form

$$f_{\mathrm{P}}(x) = x + uh(w^T x + b)$$

with non-linearity $h$ and $w, u \in \mathbb{R}^n, b \in \mathbb{R}$. They describe a plane-wise expansion or contraction of all hyperplanes orthogonal to $w$. In order to admit also non-smooth activations, we generalize the results from Rezende & Mohamed (2015) in the following theorem and obtain a sufficient criterion for the bijectivity of a planar flow $f_{\mathrm{P}}$ (see Appendix B.1 for the proof).

**Theorem 6.** *Let $f_P$ be a planar flow with activation $h$. If the one-dimensional mapping $\psi : \mathbb{R} \to \mathbb{R}$ with*

$$\psi(\lambda) = \lambda + w^T uh(\lambda)$$

*is bijective, then $f_P$ is also bijective.*

However, bijectivity is not sufficient for the planar flow to induce a density of the desired form. But similar to Theorem

*Table 1.* Conditions on the parameters of a planar flow $f_P$, such that it is a proper flow for the activation $h$ (see B.1.1 for the proofs).

| NON-LINEARITY | CONDITION |
|---|---|
| RELU | $w^T u > -1$ |
| ELU ($\alpha > 0$) | $w^T u > \max(-1, -\frac{1}{\alpha})$ |
| TANH | $w^T u \geq -1$ |
| SOFTPLUS | $w^T u > -1$ |

6, conditions can be imposed on a one-dimensional mapping such that $f_P$ describes an $\mathcal{L}$-diffeomorphism (see Appendix B.1 for the proof):

**Theorem 7.** *Let $f_P$ be a bijective planar flow. If there is a countable and closed set $N \subset \mathbb{R}$ such that the activation function $h$ is continuously differentiable on $\mathbb{R} \setminus N$ and*

$$C = \left\{ x \in \mathbb{R} \setminus N \mid 1 + w^T u h'(x) = 0 \right\}$$

*is countable, then $f_P$ is an $\mathcal{L}$-diffeomorphism. In particular, $f_P$ is a proper flow.*

Using Theorem 7, conditions for different activations such that the resulting planar flow describes a flow can be found via reducing it to a one-dimensional problem. The constraints for the most popular non-linearities are summarized in Table 1.

## 4.2. Radial Flows

Another intuitive way to perturb the identity in $\mathbb{R}^n$ is to expand or contract spherically around a centering point. This type of transformation was initially studied by Tabak & Turner (2013) and subsequently by Rezende & Mohamed (2015). These transformations of the form

$$f_R(x) = x + \beta h\big(\|x - x_0\|_2\big)(x - x_0)$$

are called *radial flows*, where $h : \mathbb{R}_+ \to \mathbb{R}_+$ is a localization function, $x_0 \in \mathbb{R}^n$ the center, and $\beta \in \mathbb{R}$. When $\beta$ is negative, a contraction occurs, and positive values lead to an expansion around the center $x_0$. The following theorem provides a sufficient criterion for the bijectivity of a radial flow if we consider non-smooth functions $h$ (see Appendix B.2 for the proof):

**Theorem 8.** *Let $f_R$ be a radial flow with localization $h$. If the one-dimensional mapping $\psi : \mathbb{R}_+ \to \mathbb{R}_+$ with*

$$\psi(r) := r + \beta h(r) r$$

*is bijective, then $f_R$ is also bijective.*

Again, bijectivity is sufficient only for the existence of the inverse which means that $f_R$, in general, does neither describe an $\mathcal{L}$-diffeomorphism nor a flow. However, this property is ensured by the conditions of the following theorem (see Appendix B.2 for the proof):

**Theorem 9.** *Let $f_R$ be a bijective radial flow. If there is a countable, closed set $N \subset \mathbb{R}_{>0}$ such that the localization function $h$ is continuously differentiable on $\mathbb{R}_{>0} \setminus N$ and*

$$C := \left\{ r \in \mathbb{R}_{>0} \setminus N \mid 1 + \beta \left( h(r) + rh'(r) = 0 \right) \right\}$$

*is countable, then $f_R$ is an $\mathcal{L}$-diffeomorphism. In particular, $f_R$ is a proper flow.*

### 4.3. Contractive Residual Flows

Contractive mappings provide a more general type of perturbations of the identity. A function $g$ is called *contractive* if there exists a constant $L < 1$ such that for all $x, y \in \mathbb{R}^n$ it holds for any norm on the vector space $\mathbb{R}^n$

$$\|g(x) - g(y)\| \leq L \|x - y\|.$$

In Behrmann et al. (2019) and Chen et al. (2019), these kinds of residual flows are called *(contractive) residual flows* and are denoted henceforth by $f_C$. On the one hand, this strong condition on $g$ gives the bijectivity of $f_C$ by the Banach's fixed point theorem; on the other hand, it follows that $f_C$ has no critical points (Behrmann, 2019, Lem. 5.4). Thus, only a few assumptions are required to guarantee that a residual flow forms an $\mathcal{L}$-diffeomorphism.

**Theorem 10.** *Let $f_C$ be a residual flow with contractive perturbation $g$. If there is a closed $\lambda^n$-null set $N \subset \mathbb{R}^n$ such that $g$ is continuously differentiable on $\mathbb{R}^n \setminus N$, then $f_C$ is an $\mathcal{L}$-diffeomorphism. In particular, $f_C$ is a proper flow.*

The proof of this statement follows directly from the property that Lipschitz continuous functions map $\lambda^n$-null sets to sets of $\lambda^n$-measure zero (Rudin, 1987, Lem. 7.25) and the fact that $\mathrm{Lip}(f_C) = 1 + L$.

## 5. Conclusion

In this paper, we have shown that the conditions on a flow need not be strictly satisfied everywhere, and a certain degree of freedom is permitted on measure-theoretically negligible sets. With this, we have justified using $\mathcal{L}$-diffeomorphism instead of a normal diffeomorphism as flows. This gain significantly increases the possibilities in the design of normalizing flows and, in particular, allows the usage of non-smooth activations. Nevertheless, we have only justified their existence and usage mathematically, far from leading to a successful practical application. Thus, future work should investigate whether and in which situations non-smooth activations provide an actual gain. Moreover, we only applied these generalizations to simple residual flows. For this reason, it remains open to what extent other flows such as more general planar flows, like Sylvester flows (Van Den Berg et al., 2018), or autoregressive flows (Huang et al., 2018; Jaini et al., 2019) also benefit from this.

## Acknowledgements

Niklas Koenen and Marvin N. Wright gratefully acknowledge the funding from the German Research Foundation (DFG) in the context of the Emmy Noether Grant 437611051.

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

## A. Generalization of the Change of Variables Formula

**Lemma A.1.** *Let $f : Z \to X$ be an $\mathcal{L}$-diffeomorphism from the measurable space $(Z, \mathcal{A}_Z)$ into the target space $(X, \mathcal{A}_X)$. Then $f$ is an $\mathcal{A}_Z$-$\mathcal{A}_X$-measurable mapping, i.e., a random variable.*

*Proof.* According to the definition of measurable functions, we have to show that every preimage of a measurable set contained in $\mathcal{A}_X$ is also an element of $\mathcal{A}_Z$. Let $B \in \mathcal{A}_X$. Since $f$ is an $\mathcal{L}$-diffeomorphism, there exist $\lambda^n$-null sets $N_Z, N_X$ such that the restriction $f : Z \setminus N_Z \to X \setminus N_X$ is bijective and continuously differentiable. We can decompose the preimage of $B$ as

$$f^{-1}(B) = \left(N_Z \cap f^{-1}(B)\right) \cup \left(Z \setminus N_Z \cap f^{-1}(B)\right).$$

Since $N_Z$ is a null set, the subset $N_Z \cap f^{-1}(B)$ also has a measure of zero. We can represent the other set of the equation above by the bijectivity of $f$ on $Z \setminus N_Z$ as

$$Z \setminus N_Z \cap f^{-1}(B) = f^{-1}(X \setminus N_X \cap B). \tag{7}$$

Thus it results from the continuity of the restricted mapping, which is in particular $\mathcal{L}(Z \setminus N_Z)$-$\mathcal{L}(X \setminus N_X)$-measurable, that the set (7) is contained in the trace $\sigma$-algebra $\mathcal{L}(Z \setminus N_Z)$, hence an element of $\mathcal{A}_Z$. In the end, we can represent $f^{-1}(B)$ as a union of a null set and a $\mathcal{A}_Z$-measurable set, which shows the claim. $\square$

***Proof of Lemma 3.*** From the Definition 2 of an $\mathcal{L}$-diffeomorphism, we obtain the existence of null sets $N_Z, N_X$ with $N_Z$ closed such that the restricted function $f : Z \setminus N_Z \to X \setminus N_X$ is bijective, continuously differentiable, and the set of critical points $C$ has measure zero. At first, we show that $C$ is a closed set. Consider the mapping $g : Z \setminus N_Z \to \mathbb{R}$ with $g(x) = \det(J_f(x))$. Because of the continuity of the determinant and the continuous differentiability of the restricted $f$, the mapping $g$ is also continuous. Therefore the preimage under $g$ of closed sets is also closed according to the topological definition of continuity, i.e.,

$$g^{-1}(\{0\}) = \{x \in Z \setminus N_Z \mid \det J_f(x) = 0\} = C$$

is a closed set; thus, $Z \setminus (N_Z \cup C)$ is open in $\mathbb{R}^n$. It follows by the inverse function theorem and its consequences (Rudin, 1976, Thm. 9.24 and 9.25) that the restriction

$$f : Z \setminus (N_Z \cup C) \to X \setminus (N_X \cup f(C))$$

forms a diffeomorphism. By assumption, $N_C$ and $C$ are $\lambda^n$-null sets, and so $N_Z \cup C$. From Sard's theorem (Milnor, 1965, §2), it follows that $f(C)$, hence especially $N_X \cup f(C)$, is a set of measure zero. $\square$

***Proof of Theorem 4.*** We obtain from Lemma 3 the existence of two null sets $N_Z, N_X$ so that the restriction $f : Z \setminus N_Z \to X \setminus N_X$ of an $\mathcal{L}$-diffeomorphism forms a diffeomorphism. Let $A \in \mathcal{A}_X$ be a measurable set in the target space $X$. Because of the additivity of $\mathbb{P}_Z$ and the definition of the pushforwad measure $\mathbb{P}_f$, it follows

$$\begin{aligned}\mathbb{P}_f(B) := \mathbb{P}_Z\left(f^{-1}(B)\right) &= \mathbb{P}_Z\left((f^{-1}(B) \setminus N_Z) \,\dot{\cup}\, (f^{-1}(B) \cap N_Z)\right) \\ &= \mathbb{P}_Z\left(f^{-1}(B) \setminus N_Z\right) + \mathbb{P}_Z\left(f^{-1}(B) \cap N_Z\right).\end{aligned} \tag{8}$$

Since $\mathbb{P}_Z$ is absolutely continuous w.r.t. the Lebesgue measure and $f^{-1}(B) \cap N_Z$ is a subset of a $\lambda^n$-null set, the right summand of term (8) vanishes. For the other term, we note the following set equality:

$$f^{-1}(B) \setminus N_Z = f^{-1}(B \setminus N_X). \tag{9}$$

Because the restriction $f : Z \setminus N_Z \to X \setminus N_X$ is a diffeomorphism and $B \setminus N_X$ is an element of the trace $\sigma$-algebra $\mathcal{L}(X \setminus N_X)$, the equation (8) can be extended by the CVF from eq. (1) as

$$\mathbb{P}_f(B) = \int_{f^{-1}(B \setminus N_X)} p_z(z) \, d\lambda^n(z) = \int_{B \setminus N_X} p_z\left(f^{-1}(x)\right) \left|\det J_{f^{-1}}(x)\right| \, d\lambda^n(x). \tag{10}$$

We define a function $p_f : X \to \mathbb{R}_+$ with

$$p_f(x) = \begin{cases} p_Z\left(f^{-1}(x)\right) \left| \det J_{f^{-1}}(x)\right| & x \in X \setminus N_X \\ 0 & x \in N_X \end{cases} \; .$$

Since $B \cap N_X \subseteq N_X$ is a $\lambda^n$-null set, we can additionally integrate in (10) over this set and obtain the expression of the distribution induced by $f$

$$\mathbb{P}_f(B) = \int_{B \setminus N_X} p_f(x)\, d\lambda^n(x) + \int_{B \cap N_X} p_f(x)\, d\lambda^n(x) = \int_B p_f(x)\, d\lambda^n(x), \tag{11}$$

hence $p_f$ forms a density of distribution $\mathbb{P}_f$. Finally, by the Radon-Nikodym theorem (Bogachev, 2006, Sec. 3.2), the density function is $\lambda^n$-unique, which shows the claim. $\qquad\square$

***Proof of Lemma 5.*** Since $f_1$ and $f_2$ are $\mathcal{L}$-diffeomorphisms, by Lemma 3 there are $\lambda^n$-null sets $N_Z^1, N_Z^2, N_X^1, N_X^2 \subset \Omega$ with $N_Z^1$ and $N_Z^2$ closed such that both

$$f_1 : \Omega \setminus N_Z^1 \to \Omega \setminus N_X^1 \qquad \text{and} \qquad f_2 : \Omega \setminus N_Z^2 \to \Omega \setminus N_X^2 \tag{12}$$

are diffeomorphisms. We remove from the domain of $f_1$ all elements mapping to the set $N_Z^2$ since $f_2$ is not a diffeomorphism on it. Hence we define

$$N_Z := N_Z^1 \cup f_1^{-1}\left(N_Z^2 \cap \Omega \setminus N_X^1\right) \qquad \text{and} \qquad N_X := N_X^2 \cup f_2\left(N_X^1 \cap \Omega \setminus N_Z^2\right). \tag{13}$$

It is relatively easy to see that both $f_1(\Omega \setminus N_Z) = \Omega \setminus (N_X^1 \cup N_Z^2)$ and $f_2(f_1(\Omega \setminus N_Z)) = \Omega \setminus N_X$ are valid. Moreover, from the continuity of $f_1$ and since $N_Z^2 \cap \Omega \setminus N_X^1$ is closed in the subspace topology on $\Omega \setminus N_X^1$, it follows that $f^{-1}(N_Z^2 \cap \Omega \setminus N_X^1)$ is closed in $\Omega \setminus N_Z^1$. Hence there exists a closed set $A \subset \Omega$ such that (see Sec. 6 in Willard, 1970 for more information)

$$f^{-1}\left(N_Z^2 \cap \Omega \setminus N_X^1\right) = A \cap \Omega \setminus N_Z^1.$$

This results with eq. (13) in

$$N_Z = N_Z^1 \cup \left(A \cap \Omega \setminus N_Z^1\right) = N_Z^1 \cup A$$

which is a closed set in $\Omega$. In summary, the mapping $f_2 \circ f_1 : \Omega \setminus N_Z \to \Omega \setminus N_X$ is well-defined, continuously differentiable, bijective, and has no critical points, since we have merely further restricted the diffeomorphisms from (12). In addition, $N_Z$ is a closed set in $\Omega$. To complete the proof, we still need to show that $N_Z$ and $N_X$ have a $\lambda^n$-measure of zero. However, this follows directly from the fact that diffeomorphisms map null sets to null sets. $\qquad\square$

The following example illustrates why we assume that the set of critical points is a null set. Indeed, one can construct bijective and continuously differentiable functions whose inverse is not differentiable on a set with positive measure.

**Example A.2.** As an example, consider the following recursively defined set on the interval $[0, 1]$ called the *Smith-Volterra-Cantor set* $C_{\text{SV}}$. The definition and properties of this set are only briefly sketched here. For more information and detailed proofs, the reader is referred to Bressoud (2008) or DiMartino & Urbina (2014). Another example can also be found in Floret (1981, Sec. 17.15).

The recursive definition of $C_{\text{SV}}$ on $[0, 1]$ starts by removing the open middle quarter from the interval. Then we create subsequent sets $S_n$ by removing an open interval of length $\frac{1}{4^n}$ from the center of each interval in $S_{n-1}$, i.e.,

$$\begin{aligned}
S_1 = & \quad \left[0, \tfrac{3}{8}\right] && \cup && \left[\tfrac{5}{8}, 1\right] \\
S_2 = & \left[0, \tfrac{5}{32}\right] \cup \left[\tfrac{7}{32}, \tfrac{3}{8}\right] \cup \left[\tfrac{5}{8}, \tfrac{25}{32}\right] \cup \left[\tfrac{27}{32}, 1\right] \\
S_3 = & \left[0, \tfrac{1}{16}\right] \cup \left[\tfrac{3}{32}, \tfrac{5}{32}\right] \cup \left[\tfrac{7}{32}, \tfrac{9}{32}\right] \cup \left[\tfrac{5}{16}, \tfrac{3}{8}\right] \cup \left[\tfrac{5}{8}, \tfrac{11}{16}\right] \cup \left[\tfrac{23}{32}, \tfrac{25}{32}\right] \cup \left[\tfrac{27}{32}, \tfrac{29}{32}\right] \cup \left[\tfrac{15}{16}, 1\right] \\
& \;\vdots
\end{aligned}$$

Figure 1. Visualization of $S_n$.

Now the Smith-Volterra-Cantor set is defined as the following intersection of all $S_n$

$$C_{\text{SV}} := \bigcap_{n=1}^{\infty} S_n,$$

which is closed because of the countable intersection of closed sets. In addition, in each recursion step in each of the $2^{n-1}$ intervals, the middle pieces with the Lebesgue measure of $\frac{1}{4^n}$ are removed. Hence it holds

$$\lambda^1(C_{\text{SV}}) = 1 - \sum_{n=1}^{\infty} \frac{2^{n-1}}{4^n} = 1 - \frac{1}{4} \sum_{n=0}^{\infty} \left(\frac{1}{2}\right)^n = 1 - \frac{1}{2} = \frac{1}{2}$$

due to the geometric series. In order to construct a counterexample, we consider the function $d : [0,1] \rightarrow \mathbb{R}$ with $d(x) = \inf_{c \in C_{\text{SV}}} |x - c|$. This function is obviously continuous and $d(x) = 0$ holds if and only if $x \in C_{\text{SV}}$, otherwise it only takes positive values. Because of the continuity, this function is integrable and it follows from the fundamental theorem of calculus that

$$f : (0,1) \rightarrow f\big((0,1)\big)$$
$$f(x) = \int_0^x d(z) \, dz,$$

is continuously differentiable. Moreover, $f$ is injective resulting from the construction of $C_{\text{SV}}$ and the fact that $d$ is not constant zero on any interval with positive measure. But the set of critical points of $f$ is the Smith-Volterra-Cantor set, which is not a $\lambda$-null set.

## B. Non-smooth Activations in Residual Flows

### B.1. Planar Flows

Theorem 6 presented here corresponds to a generalization of the proof given by Rezende & Mohamed (2015) since not only the smooth activation function $h(x) = \tanh(x)$ is considered. However, the argument in the proof is very similar.

***Proof of Theorem 6.*** Let $y \in \mathbb{R}^n$ be arbitrary. Then we have to show that the following equation has a unique solution

$$f_{\text{P}}(x) = x + uh\left(w^T x + b\right) = y. \tag{14}$$

If $w = 0$ holds, then a unique solution is given by $x = y - uh(b)$, wherefrom bijectivity results. For this reason, let $w \neq 0$ in the following, thus $w$ spans a one-dimensional linear subspace $W$ in $\mathbb{R}^n$. Consequently, each element $x \in \mathbb{R}^n$ has a unique orthogonal decomposition $x = x_{\parallel} + x_{\perp}$ with $x_{\parallel} \in W$ and $x_{\perp} \in W^{\perp} := \{x \in \mathbb{R}^n \mid w^T x = 0\}$. Due to the orthogonality of $x_{\perp}$ and $w$, the following solution of the orthogonal component depending on the parallel one can be inferred from eq. (14)

$$x_{\perp} = y - x_{\parallel} - uh\left(w^T x_{\parallel} + b\right). \tag{15}$$

Since $W$ is a one-dimensional linear subspace, there is a unique $\lambda \in \mathbb{R}$ with $x_{\parallel} = w \frac{\lambda}{w^T w}$. By using this representation, the original equation multiplied by $w^T$ from the left yields

$$w^T y = w^T x_{\perp} + w^T w \frac{\lambda}{w^T w} + w^T uh(\lambda + b) = \lambda + w^T uh(\lambda + b) \tag{16}$$
$$= \psi(\lambda + b) - b,$$

where the last equation in (16) follows again by $w^T x_{\perp} = 0$. The assumed bijectivity of $\psi$ leads to the unique existence of a $\lambda_y \in \mathbb{R}$, which solves the equation above. In addition, this implies the existences of $x_{\parallel}$ and $x_{\perp}$, thus

$$x = x_{\parallel} + x_{\perp} = y - uh\left(\lambda_y + b\right)$$

is the unique solution of the initial equation (14) and consequently the bijectivity of $f_{\text{P}}$ follows. $\qquad \square$

***Proof of Theorem 7.*** In case $w = 0$, $f_{\text{P}}$ represents an affine linear mapping describing a diffeomorphism, hence an $\mathcal{L}$-diffeomorphism. For this reason, let $w \neq 0$. Consequently, $w$ spans a one-dimensional linear subspace $W$, from

which follows a unique orthogonal decomposition of the vector space $\mathbb{R}^n = W + W^\perp$ as a direct sum. This leads to a characterization of the vector space as a disjoint union of hyperplanes, i.e.,

$$\mathbb{R}^n = \dot{\bigcup_{\lambda \in \mathbb{R}}} H_\lambda \qquad \text{with} \qquad H_\lambda := \left\{ w \frac{\lambda}{w^T w} + x_\perp \,\middle|\, x_\perp \in W^\perp \right\}.$$

Consider the function $\tau : \mathbb{R}^n \to \mathbb{R}$ with $\tau(x) = w^T x + b$ mapping every element of a hyperplane $H_\lambda$ to the same value $\lambda + b$. Since the activation function $h$ is not continuously differentiable on the countable set $N$, we remove all hyperplanes mapping to $N$ under $\tau$. So we define

$$H_N := \bigcup_{n \in N} H_{n-b} \qquad \text{such that} \qquad \tau(H_N) = N.$$

Because $\tau$ is continuous and $N$ is closed in $\mathbb{R}$, it follows that $H_N = \tau^{-1}(N)$ is closed in $\mathbb{R}^n$. Moreover, $H_N$ as a countable union of hyperplanes is also a null set w.r.t. the Lebesgue measure $\lambda^n$ due to the subadditivity. Accordingly, the restriction $f_P : \mathbb{R}^n \setminus H_N \to \mathbb{R}^n \setminus f_P(H_N)$ is bijective and continuously differentiable. Furthermore, $H_N$ is a closed $\lambda^n$-null set. We note that the image of a hyperplane $H_\lambda$ under $f_P$ is also a hyperplane; hence $f_P(H_N)$ forms a countable union of hyperplanes which is a null set. Finally, it is left to show that the set of critical points of the restricted planar flow is also a null set. By the matrix determinant lemma we get for the set of critical points the set equality

$$C_{f_P} := \{ x \in \mathbb{R}^n \setminus H_N \,|\det J_{f_P}(x) = 0 \} = \left\{ x \in \mathbb{R}^n \setminus H_N \,\middle|\, 1 + w^T u h'(\tau(x)) = 0 \right\}.$$

This gives $\tau(C_{f_P}) = C$, which we have assumed to be a countable set. Hence

$$C_{f_P} = \tau^{-1}(C) = \bigcup_{c \in C} \tau^{-1}(\{c\}) = \bigcup_{c \in C} H_{c-b},$$

which is as a countable union of hyperplanes a $\lambda^n$-null set. □

### B.1.1. EXAMPLES FOR SOME ACTIVATIONS

In the following, we infer conditions for particular choices of activation functions for a planar flow $f_P$. A visualization of the crucial function for each proof can be found in Figure 2, where the conditions on the flow are fulfilled, just (not) fulfilled and not fulfilled anymore.

**Lemma B.1** (ReLU). *If $w^T u > -1$ holds, then $f_P$ with activation ReLU is an $\mathcal{L}$-diffeomorphism.*

*Proof.* Consider the mapping $\psi : \mathbb{R} \to \mathbb{R}$ given by

$$\psi(\lambda) = \lambda + w^T u \, \text{ReLU}(\lambda + b) = \begin{cases} \lambda & , \lambda < -b \\ \lambda + w^T u (\lambda + b) & , \lambda \geq -b \end{cases}.$$

Because of the inequality $w^T u > -1$, this mapping is strictly monotonically increasing and thus obviously bijective. Consequently, the bijectivity of the planar flow $f_P$ follows from Theorem 6. Furthermore, the ReLU activation is continuously differentiable on $\mathbb{R} \setminus \{0\}$ and it holds for all $\lambda \in \mathbb{R} \setminus \{0\}$

$$1 + w^T u \, \text{ReLU}'(\lambda) = \begin{cases} 1 & , \lambda < 0 \\ 1 + w^T u & , \lambda > 0 \end{cases} \neq 0.$$

Therefore, the planar flow $f_P$ with activation ReLU has no critical points; thus, the claim follows from Theorem 7. □

**Lemma B.2** (tanh). *If $w^T u \geq -1$ holds, then $f_P$ with activation $\tanh$ is an $\mathcal{L}$-diffeomorphism.*

*Proof.* Consider the function $\psi : \mathbb{R} \to \mathbb{R}$ given by $\psi(\lambda) = \lambda + w^T u \tanh(\lambda)$ with derivative $\psi'(\lambda) = 1 + \frac{w^T u}{\cosh(\lambda)^2}$. Since the hyperbolic cosine is 1 only at 0 and otherwise always greater than 1, we get with the assumed inequality $\psi'(\lambda) \geq 0$ and equality only if $\lambda = 0$ and $w^T u = -1$. Therefore, the function $\psi$ is strictly monotonically increasing, hence injective. Moreover, the surjectivity follows from the boundedness of the hyperbolic tangent. Consequently, the bijectivity of the planar flow $f_P$ follows from Theorem 6. The activation function is continuously differentiable, and as seen earlier, the equation $1 + w^T u \tanh(\lambda) = 0$ is only satisfied if $w^T u = -1$ and $\lambda = 0$. In any case, the set of critical points is a countable set, so the claim follows from Theorem 7. □

**Lemma B.3** (ELU). *If $w^T u > \max(-1, -\frac{1}{\alpha})$ holds, then $f_P$ with activation ELU ($\alpha > 0$) is an $\mathcal{L}$-diffeomorphism.*

*Proof.* Consider the function $\psi : \mathbb{R} \to \mathbb{R}$ with

$$\psi(\lambda) = \lambda + w^T u \, \text{ELU}(\lambda + b) = \begin{cases} \lambda + w^T u \alpha \left( e^{\lambda+b} - 1 \right) & , \lambda \leq -b \\ \lambda + w^T u (\lambda + b) & , \lambda > -b \end{cases} .$$

This function is continuously differentiable on $\mathbb{R} \setminus \{b\}$ with derivative given by

$$\psi'(\lambda) = \begin{cases} 1 + w^T u \alpha e^{\lambda+b} & , \lambda < -b \\ 1 + w^T u & , \lambda > -b \end{cases} .$$

By the assumed inequality and $e^{\lambda+b} \in (0,1)$ for $\lambda < -b$, we obtain for $\lambda < -b$

$$\psi'(\lambda) = 1 + w^T u \alpha e^{\lambda+b} > 1 + \max\left(-1, -\frac{1}{\alpha}\right) \alpha e^{\lambda+b} > 1 + \max(-\alpha, -1) \geq 0. \tag{17}$$

In addition, we get the positivity for the other case $\lambda > -b$

$$\psi'(\lambda) = 1 + w^T u > 1 + \max\left(-1, -\frac{1}{\alpha}\right) \geq 0. \tag{18}$$

Because of the limit $\lim_{\lambda \searrow -b} \psi(\lambda) = \psi(-b) = -b$, the function $\psi$ is strictly monotonically increasing, thus injective. Furthermore, surjectivity results from the mean value theorem; hence $f_P$ is bijective by Theorem 6. Additionally, the activation function ELU is continuously differentiable on $\mathbb{R} \setminus \{b\}$, and similar to (17) and (18) the inequality $\psi'(\lambda) = 1 + w^T u \text{ELU}(\lambda) > 0$ follows for all $\lambda \neq 0$. Therefore we can apply Theorem 7, which shows the claim. $\square$

**Lemma B.4** (Softplus). *If $w^T u > -1$ holds, then $f_P$ with activation Softplus is an $\mathcal{L}$-diffeomorphism.*

*Proof.* For the function $\psi : \mathbb{R} \to \mathbb{R}$ given by $\psi(\lambda) = \lambda + w^T u \text{Softplus}(\lambda + b)$ with $\text{Softplus}(\lambda) = \log\left(1 + e^{\lambda+b}\right)$ the derivative is

$$\psi'(\lambda) = 1 + w^T u \frac{e^{\lambda+b}}{1 + e^{\lambda+b}} = 1 + w^T u \frac{1}{1 + e^{-(\lambda+b)}}.$$

Since the range of the factor $\frac{1}{1+e^{-(\lambda+b)}}$ is in the interval $(0, 1)$, the assumed inequality results for all $\lambda \in \mathbb{R}$ in

$$\psi'(\lambda) > 1 - \frac{1}{1 + e^{-(\lambda+b)}} > 0. \tag{19}$$

Consequently, the function $\psi$ is strictly monotonically increasing, hence injective. By continuity, the limit $\lim_{\lambda \to -\infty} = -\infty$ results. Without loss of generality, we consider $\lambda > -b$ in the following. In this case, the inequality $\log(1 + e^{\lambda+b}) \leq \log(2e^{\lambda+b})$ holds, and we obtain for $w^T u < 0$

$$\lim_{\lambda \to \infty} \psi(\lambda) \geq \lim_{\lambda \to \infty} \lambda + w^T u \log\left(2e^{x+b}\right) = \lim_{\lambda \to \infty} \lambda + w^T u \log(2) (\lambda + b) = \infty.$$

For the other case $w^T u \geq 0$, results $\lim_{\lambda \to \infty} \psi(\lambda) \geq \lim_{\lambda \to \infty} \lambda = \infty$. Thus, the surjectivity follows from the mean value theorem. Therefore, we get the bijectivity from Theorem 6. In addition, the activation function Softplus is everywhere continuously differentiable and has not critical points, so the claim follows from Theorem 7. $\square$

### B.2. Radial Flows

Theorem 8 presented here corresponds to a generalization of the proof given by Rezende & Mohamed (2015), since not only the localization function $h(r) = \frac{1}{\alpha+r}$ is considered. However, the arguments in the proof are very similar.

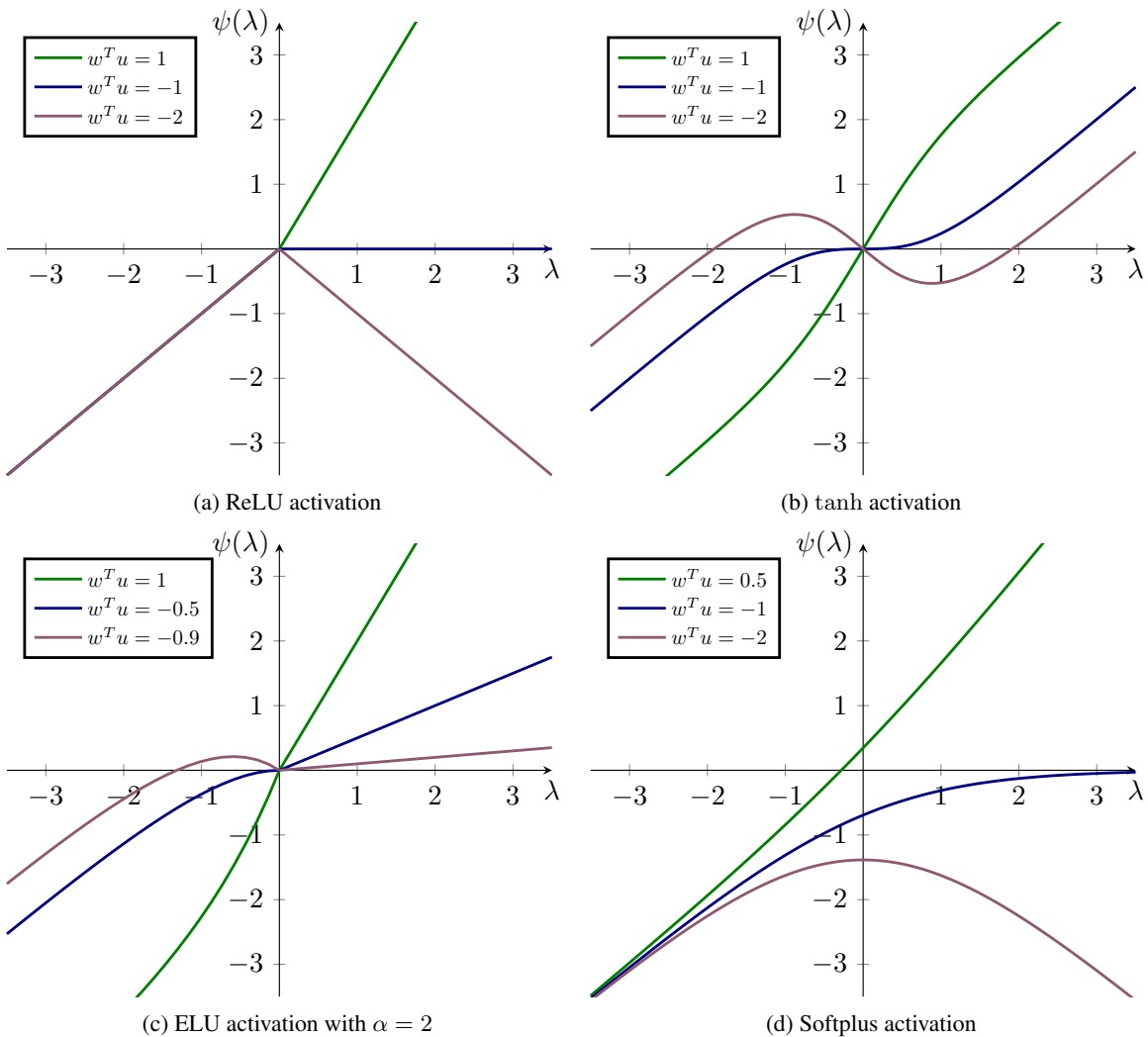

(a) ReLU activation

(b) tanh activation

(c) ELU activation with $\alpha = 2$

(d) Softplus activation

*Figure 2.* Visualization of the function $\psi$ considered in the proofs of Section B.1.1. For each of the activations ReLU, tanh, ELU$(\alpha = 2)$ and Softplus, we plotted the function $\psi$ with $b = 0$ for three different choices of $w^T u$; the conditions for an $\mathcal{L}$-diffeomorphism are satisfied (green), just (not) satisfied (blue), and not satisfied (magenta).

***Proof of Theorem 8.*** Let $y \in \mathbb{R}^n$ arbitrary. We have to show for bijectivity that the following equation has a unique solution

$$f_{\mathrm{R}}(x) = x + \beta h(r)(x - x_0) = y \qquad \text{where} \qquad r := \|x - x_0\|_2. \tag{20}$$

If $y = x_0$, we then obtain the equality

$$0 = \|y - x_0\|_2 = \|x - x_0 + \beta h(r)(x - x_0)\|_2 = r + \beta h(r)r = \psi(r)$$

after rearranging equation (20) and taking the Euclidean norm. Since $\psi(0) = 0$ and $\psi$ was assumed to be bijective, it follows that $\|x - x_0\| = 0$ which is equivalent to $x = x_0$ because of the definiteness of the Euclidean distance. Hereafter let $y \in \mathbb{R}^n \setminus \{x_0\}$, i.e., $r = \|x - x_0\|_2 > 0$. In this case, each element $x - x_0$ with $x \in \mathbb{R}^n \setminus \{x_0\}$ can be expressed unambiguously as the product of its projection on the unit sphere and its Euclidean distance. Thus, for $x \in \mathbb{R}^n \setminus \{x_0\}$ there exists a unique normalized direction vector $\hat{x} \in S_1(x_0) := \{x \in \mathbb{R}^n \mid \|x - x_0\|_2 = 1\}$ such that $x = x_0 + r\hat{x}$ where $r := \|x - x_0\|_2$. Substituting this representation into equation (20), one obtains after conversion

$$y - x_0 = \hat{x} \left( r + \beta h(r)r \right) = \hat{x}\psi(r), \tag{21}$$

and by taking of the norm of this

$$0 < \|y - x_0\|_2 = \|\hat{x}\psi(r)\|_2 = \psi(r).$$

Because of the bijectivity of $\psi$, the unique existence of a radius $r_y > 0$ around the centering point results. Due to the fact that $\psi(r) > 0$ remains valid for $r > 0$, equation (21) gives the following unambiguous expression of the direction vector $\hat{x}$

$$\hat{x}_y := \frac{y - x_0}{\psi(r_y)} = \frac{y - x_0}{r_y + \beta h(r_y)r_y}.$$

Hence $x = x_0 + r_y\hat{x}_y$ is the unique solution of the original equation (20), concluding finally that $f_R$ is bijective. $\quad\square$

*Proof of Theorem 9*. According to the requirement, the localization function is not continuously differentiable for all radii, so spheres with such distances around the centering point must be removed from the domain of $f_R$. Furthermore, the point $x_0$ corresponding to a radius of $0$ must be eliminated in order to restrict the localization function to an open set, thus allowing us to verily speak of differentiability. For this purpose we define

$$S := \bigcup_{r \in N} S_r(x_0) \cup \{x_0\} \qquad \text{where} \qquad S_r(x_0) := \{x \in \mathbb{R}^n \mid \|x - x_0\|_2 = r\}.$$

Since the shifted Euclidean norm $\tau(x) := \|x - x_0\|_2$ is continuous, it follows that the preimage of the closed set $N \cup \{0\}$ is also closed. Therefore, the set of eliminated points

$$S = \bigcup_{r \in N} \tau^{-1}(\{r\}) \cup \tau^{-1}(\{0\}) = \tau^{-1}(N \cup \{0\})$$

is closed; moreover, it is a set of measure zero because a countable union of spheres and points is a $\lambda^n$-null set. In summary, the restriction $f_R : \mathbb{R}^n \setminus S \to \mathbb{R}^n \setminus f_R(S)$ describes a bijective and continuously differentiable mapping. Besides, the following is valid for every $x \in S_r(x_0)$

$$\|f_R(x) - x_0\|_2 = \|x - x_0 + \beta h(r)(x - x_0)\|_2 = r + \beta h(r)r. \tag{22}$$

This indicates that $f_R$ maps spheres with radius $r$ to spheres with radius $r + \beta h(r)r$ around the centering point $x_0$; thus, $f_R(S)$ is also a countable union of null sets, therefore, itself a null set. Finally, it remains to show that the set of critical points of this restriction is also a Lebesgue null set. From the matrix determinant lemma and the higher dimensional differentiation rules, the Jacobian-determinant of $f_R$ at position $x \in \mathbb{R}^n \setminus S$ is given by

$$\det(J_{f_R}(x)) = (1 + \beta h(r))^{n-1}(1 + \beta h(r) + \beta h'(r)r). \tag{23}$$

Since $f_R(x_0) = x_0$ holds and $f_R$ is bijective, both $\|f_R(x) - x_0\|_2 > 0$ and $r = \|x - x_0\|_2 > 0$ hold for all $x \in \mathbb{R}^n \setminus S$. Thus equation (22) yields

$$\frac{\|f_R(x) - x_0\|_2}{r} = 1 + \beta h(r) > 0.$$

Knowing that this term does not vanish for any $x \in \mathbb{R}^n \setminus S$, the set of critical points can be represented using equation (23) like

$$C_{f_R} := \{x \in \mathbb{R}^n \setminus S \mid \det J_{f_R}(x) = 0\} = \{x \in \mathbb{R}^n \setminus S \mid 1 + \beta h(\tau(x)) + \beta h'(\tau(x)) = 0\}.$$

This gives $\tau(C_{f_R}) = C$, which we have assumed to be a countable set. Hence

$$C_{f_R} = \tau^{-1}(C) = \bigcup_{c \in C} \tau^{-1}(\{c\}) = \bigcup_{c \in C} S_c(x_0),$$

which is as a countable union of spheres a $\lambda^n$-null set. $\quad\square$