# OpenReview forum: "Generalization of the Change of Variables Formula with Applications to Residual Flows"
_ICML.cc/2021/Workshop/INNF — INNF+ 2021 poster_

### Official Review · Reviewer_gaDn · 2021-06-11

**Rating:** Borderline Accept
**Confidence:** 4

**Summary:**

The authors of the paper take a closer look at the change of variables formula that is the central component of normalizing flows (or, broadly speaking, flow-based models). The authors show conditions of a flow to be properly defined. These are important and path a way for future research. However, there is still much to do and it is rather unclear whether the presented ideas are of practical use.

**Justification For Rating:**

Strengths:
S1: The paper has a theoretical character, thus, it is universal.
S2: The presented ideas are applied to some of the existing flows, e.g., planar and radial flows.

Weaknesses:
W1: The presented theory is applied to rather simple flows. As mentioned in the conclusion, residual flows and Sylvester flows seem to be more interesting to analyze.
W2: It is still unclear whether the presented ideas are of real practical use.

---

### Official Review · Reviewer_BcaD · 2021-06-11

**Rating:** Accept
**Confidence:** 3

**Summary:**

This paper proposes to relax the requirements on functions used to construct flow-based models: rather than requiring them to be bijective and continuously differentiable everywhere, this can instead be required "almost everywhere", i.e. it is okay for this condition to be violated on sets of measure zero. This is particularly relevant because modern neural networks often use non-smooth nonlinearities that give rise to such functions (e.g. ReLU). Put differently, the paper provides the theoretical underpinnings necessary for flow-based models with non-smooth nonlinearities.

**Justification For Rating:**

While I am definitely not the best person to assess the validity of the theorems and proofs in detail, the theory looks solid as far as I can tell, and this work provides a useful formalism for reasoning about a more general class of flow-based models.

The developed theory is applied to several existing types of flows (planar, radial, contractive residual), though no experimental results are provided. While I would personally have liked to see some empirical evidence (even if it is just on toy data), I appreciate that might not be the goal of the paper -- and of course, e.g. Chen et al. have already shown that this works in practice for contractive residual flows.

---

### Decision · Program_Chairs · 2021-06-14

Accept (poster)